# FedSDR: Federated Graph Learning with Structural Noise Detection and Reconstruction

## Abstract

Federated Graph Learning (FGL) has emerged as a principled framework for decentralized training of Graph Neural Networks (GNNs) while preserving data privacy. In subgraph-FL scenarios, however, structural noise arising from data collection and storage can damage the GNN message-passing scheme of clients, leading to conflicts in collaboration. Existing approaches exhibit two critical limitations: 1) Globally, they fail to identify corrupted clients, causing destructive message-passing conflicts. 2) Locally, the global GNN performs poorly on these clients due to structural noise, limiting their ability to benefit from federated collaboration. To address these challenges, we propose **FedSDR**, a robust FGL framework against high-structural-noise scenarios. Specifically, SNAA introduces a noise evaluation metric to detect corrupted clients and reduce their contributions, thereby mitigating the impact of noise on the global GNN. Furthermore, RLSR leverages the knowledge from the healthy global model to repair locally corrupted graph structures. Extensive experiments demonstrate that FedSDR outperforms state-of-the-art methods across various scenarios under structural noise.

## 1 Introduction

Graph Neural Networks (GNNs) Scarselli et al. (2008); Shchur et al. (2018b); Luan et al. (2022); Zhu et al. (2021) have emerged as a powerful framework for learning from graph-structured data, leveraging message-passing mechanisms to capture complex relational patterns. However, traditional GNNs training paradigms rely on centralized access to the entire graph Wu et al. (2022; 2021), which becomes impractical in the real world due to privacy considerations Zhang et al. (2024). This limitation is particularly acute in domains such as healthcare Li et al. (2023), finance Schreyer et al. (2022), and social networks Zhang et al. (2024). To address these challenges, recent advances have integrated Federated Learning (FL) McMahan et al. (2017b); Li et al. (2020a); Huang et al. (2022) with GNNs, giving rise to Federated Graph Learning (FGL) Tan et al. (2024); Wan et al. (2024); Zhang et al. (2021b). FGL extends FL principles to decentralized graph data, enabling clients to collaboratively train GNNs while preserving their privacy.

Although existing FGL frameworks make preliminary attempts to address structural heterogeneity and domain shift, they exhibit significant limitations when confronted with substantial structural noise—corruptions in the graph topology, such as missing edges and spurious edges. Such noise is prevalent in real-world graph applications. For instance, in credit card fraud detection GNNs, fraudsters may create transactions with a few high-credit users to disguise themselves, thus evading detection Jin et al. (2020). Similarly, social networks like Facebook or LinkedIn frequently exhibit noise from inauthentic follower relationships Dai et al. (2018). These malicious transactions and disguised relationships represent forms of edge corruption. Furthermore, results in Fig. 1 demonstrate that structurally corrupted graphs severely degrade model performance. Consequently, these limitations reveal two critical deficiencies in practical deployment scenarios as outlined below.

For global collaboration, structural noise can damage the GNN message-passing scheme of clients. Therefore, structurally corrupted clients often impose harmful knowledge on the global GNN, introducing destructive global knowledge inconsistencies. Existing methods rarely address the challenge arising from structural noise. Specifically, Li et al. (2024b) only attempts to mitigate label noise by filtering noisy nodes. Similarly, another method Fu et al. (2024) indirectly measures neighbor information deviation through statistical indicators. Neither method effectively identifies

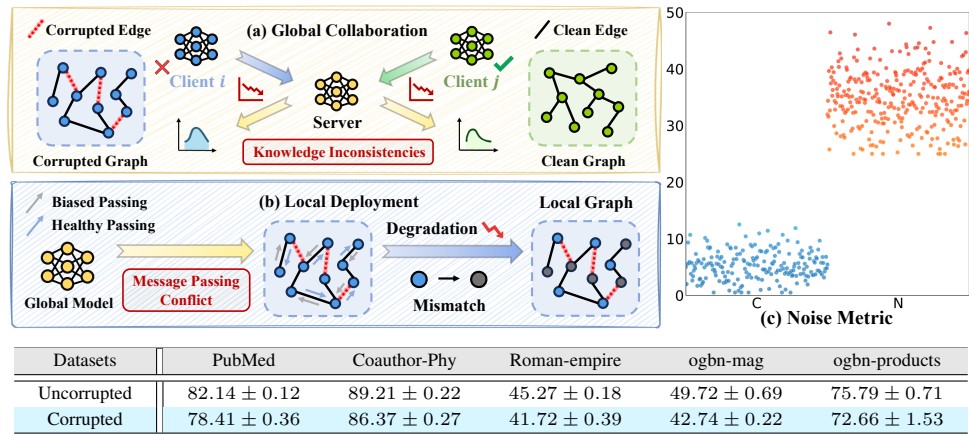

| Datasets | PubMed | Coauthor-Phy | Roman-empire | ogbn-mag | ogbn-products |
|---|---|---|---|---|---|
| Uncorrupted | $82.14 \pm 0.12$ | $89.21 \pm 0.22$ | $45.27 \pm 0.18$ | $49.72 \pm 0.69$ | $75.79 \pm 0.71$ |
| Corrupted | $78.41 \pm 0.36$ | $86.37 \pm 0.27$ | $41.72 \pm 0.39$ | $42.74 \pm 0.22$ | $72.66 \pm 1.53$ |

Figure 1: **Problem illustration.** We illustrate the challenges under high-structural-noise scenarios. (a) Globally, structural corruption introduces **harmful knowledge**, causing **knowledge inconsistencies** that undermine collaborative learning. (b) Locally, mitigating contributions of corrupted clients **biases** their message-passing schemes, creating **feature mismatches** that degrade global model performance on them. (c) Spectral metric is significantly lower for corrupted clients (C) than for clean ones (N), which enables reliable noise detection. The table below compares the average test accuracy of models trained with corrupted and uncorrupted graphs under FedAvg, with a corruption ratio of 1 and a noise extent of 0.5. Implementation details are provided in Sec. 5.1.

structural noise, leading to performance degradation in structural noise scenarios. The key to addressing structural noise is identifying structurally corrupted clients and reducing their impact. This observation raises a key research question: 1) *How can we **detect** and **mitigate the adverse impact of corrupted clients** under high-structural-noise scenarios?*

For local deployment, however, simply mitigating the impact of corrupted clients is not a viable solution, introducing two critical limitations. First, it discards potentially valuable knowledge from these clients, as even corrupted data may contain partial valid patterns that could benefit the global model. More importantly, such a mitigation creates a fundamental mismatch during local deployment. The global model, trained on clean client data in large part, suffers severe performance degradation on corrupted clients. Therefore, it denies them any benefits of collaborative learning. A more fundamental solution is to directly repair the corrupted graph structure locally. This approach maintains the participation of all clients while avoiding the above two flaws. This analysis triggers the following question: 2) *How to design an effective method to **repair corrupted graph structure** without compromising valuable knowledge during aggregation?*

To address the first challenge, motivated by the relationship between graph structure and spectral properties Tan et al. (2024); Kreuzer et al. (2021); Bo et al. (2023a), we hypothesize that structural noise will manifest as detectable anomalies in the spectral domain. Moreover, we develop an innovative spectral fidelity evaluation metric accordingly. To validate this, we first conduct experiments revealing that structurally corrupted clients exhibit a significantly lower evaluation metric than clean ones (Fig. 1). Building on this insight, we introduce spectra-guided **Structural Noise-Aware Aggregation** (SNAA). This principled method detects structurally corrupted clients and reduces their contributions through a weighting mechanism based on their extent of corruption. By dynamically redistributing client contributions, SNAA effectively mitigates the adverse impact of corrupted clients, fostering resilient and adaptive collaborative learning in the presence of structural noise.

To tackle the second issue, we develop a graph repair strategy leveraging global knowledge derived from robust SNAA aggregation. Since SNAA inherently establishes a reliable global message-passing scheme, it synthesizes healthy global knowledge that suggests consensus among uncorrupted clients. This consensus is strongly reflected in the feature similarity, which reflects semantic validity Jin et al. (2021). Therefore, corrupted connections exhibit significant deviations in feature similarity against this knowledge. Motivated by this, we propose Robust Local Structure Reconstruction (RLSR), which leverages the knowledge of the healthy global model to guide local graph repair. In contrast to removing the corrupted structure, our approach selectively reconnects edges based on feature similarity alignment with the global model, while maintaining valid local structure patterns. Moreover, it corrects the intrinsic spectral properties of the graph (Fig. 2), which are critical for effective GNN

message-passing schemes Stachenfeld et al. (2020); Geisler et al. (2024); Balcilar et al. (2021a). Therefore, the repaired structure remains consistent with collaboratively learned knowledge and enhances robust GNN training. Combining both strategies for noise-aware aggregation and structure reconstruction, we propose FedSDR, a novel framework Federated Graph Learning with Structural noise Detection and Reconstruction. Our principal contributions are summarized as follows.

- We are the first to reveal that structurally corrupted clients manifest as spectral outliers, and that corrupted connections introduce observable discrepancies in feature similarities between the local and global models.

- We propose FedSDR, an innovative framework that detects structurally corrupted clients from a spectral perspective and mitigates their adverse impact through reweighting and structure reconstruction.

- We conducted extensive experiments in high-structural-noise settings to demonstrate that FedSDR outperforms state-of-the-art methods, achieving superior robustness and accuracy.

## 2 RELATED WORK

### 2.1 GRAPH NEURAL NETWORKS

Graph Neural Networks (GNNs) Wu et al. (2020); Zhou et al. (2020); Zhang et al. (2019) have emerged as a powerful framework for learning representations of graph-structured data, broadly categorized into spectral and spatial approaches. Spectral GNNs Bo et al. (2023b); Balcilar et al. (2021b) design filters in the spectral domain, with early methods such as ChebNet Defferrard et al. (2016) employing fixed polynomial approximations and more recent advances, for instance, BernNet He et al. (2021) and JacobiConv Wang & Zhang (2022), introducing scalable spectral filters for enhanced flexibility. In contrast, spatial GNNs You et al. (2020); Bui et al. (2022) operate via direct neighborhood aggregation, with influential examples including GIN Xu et al. (2019), GAT Veličković et al. (2017), and GraphSAGE Hamilton et al. (2017). While spatial methods excel in scalability and inductive learning, they can face challenges in handling structural perturbation and long-range dependencies, issues that spectral methods may mitigate through global frequency-aware filtering. This motivates our work to integrate spectral robustness into federated GNNs, bridging a gap in the existing literature where spectral approaches remain underexplored within decentralized settings.

### 2.2 ROBUST FEDERATED LEARNING

Robust Federated Learning seeks to address the challenges posed by data heterogeneity, adversarial attacks, and model inconsistency in distributed collaborative training. Traditional approaches like FedAvg McMahan et al. (2017a) often fail to handle non-IID data or malicious clients, prompting advancements in both Byzantine robustness and adversarial resilience. For instance, Zhu et al. (2023) introduces a high-dimensional robust aggregation protocol with near-optimal statistical rates, while Zhang et al. (2023) proposes DBFAT to balance clean and robust accuracy through decision-boundary optimization under non-IID settings. For heterogeneous scenarios, Huang et al. (2024a) utilizes public data entropy to filter malicious updates, whereas Fang & Ye (2022b) handles the label noise via a robust noise-tolerant loss function. Nevertheless, these methods lack specialized mechanisms to handle structural corruption in decentralized graph data, limiting their effectiveness in real-world scenarios where such challenges are prevalent.

### 2.3 FEDERATED GRAPH LEARNING

Federated Graph Learning (FGL) enables collaborative training of GNNs across decentralized clients while preserving graph data privacy Wan et al. (2024); Liu & Yu (2022); Huang et al. (2024c); Chen et al. (2024). Current research in FGL primarily explores two paradigms: inter-graph and intra-graph learning. In inter-graph FGL, clients train GNNs on distinct local graphs to achieve globally generalizable models or enhance local performance through federated collaboration Huang et al. (2024b); Tan et al. (2023). Intra-graph FGL focuses on tasks like node classification Huang et al. (2023); Li et al. (2024d), link prediction Li et al. (2024a), and community detection within shared or partitioned graphs Leeney & McConville (2023); Baek et al. (2023). However, existing

methods fail to account for client-specific structural noise. This paper primarily focuses on intra-graph FGL scenarios under high-structural-noise scenarios, innovatively revealing the unexplored impacts of structural corruption. We effectively address the issue from two aspects: detecting structurally corrupted clients through spectral analysis and repairing local graph structures by aligning them with global feature similarities, thereby enhancing both global aggregation and local model performance.

## 3 PRELIMINARY

### 3.1 STRUCTURAL FIDELITY EVALUATION METRIC

Consider the undirected graph represented as $\mathcal{G} = (\mathcal{V}, \mathcal{E})$, where $\mathcal{V}$ represents the collection of nodes with a total of $|\mathcal{V}| = N$ vertices, and $\mathcal{E} \subseteq \mathcal{V} \times \mathcal{V}$ defines the set of edges that connect pairs of nodes. The graph corresponds to an adjacency matrix $\mathbf{A} \in \{0, 1\}^{N \times N}$, where $\mathbf{A}_{uv} = 1$ represents the existing edge $e_{uv} \in \mathcal{E}$ and $\mathbf{A}_{uv} = 0$ otherwise. The degree matrix is constructed as $\mathbf{D} = \mathrm{diag}(d_1, ..., d_N)$, where each $d_i = \sum_{j=1}^{N} \mathbf{A}_{ij}$ represents the degree of node $i \in \mathcal{V}$, with the assertion $d_i > 0$. Then the Laplacian matrix is defined as $\mathbf{L} = \mathbf{D} - \mathbf{A}$. Our $\underline{s}$tructural $\underline{fide}$lity evaluation metric $S_{\mathrm{ide}}$ derives from spectral graph theory, as the eigenvalues of the Laplacian encode graph topology Fiedler (1973); Von Luxburg (2007). It is defined as:

$$S_{\mathrm{ide}} = \frac{\mathbf{D}^T \mathbf{L} \mathbf{D}}{\mathbf{D}^T \mathbf{D}}, \tag{1}$$

where the Laplacian quadratic form $\mathbf{D}^T \mathbf{L} \mathbf{D}$ quantifies the degree-weighted connectivity disruptions, while the denominator $\mathbf{D}^T \mathbf{D}$ provides normalization. Structural noise increases connectivity disruptions, which in turn decreases the numerator. This causal relationship ensures that corrupted clients yield a significantly lower $S_{\mathrm{ide}}$ value, enabling reliable noise detection and client reweighting.

### 3.2 FEDERATED LEARNING

Federated learning establishes a decentralized optimization paradigm where $K$ distributed clients collaboratively train machine learning models while maintaining data locality. In federated optimization, we consider a global model with parameters $\theta \in \mathbb{R}^d$, where $d$ denotes the parameter dimension. The collaborative objective minimizes:

$$\min_\theta \mathcal{F}(\theta) = \min_\theta \sum_{k=1}^{K} w_k \mathcal{L}_k(\theta), \tag{2}$$

where $\mathcal{F}(\theta)$ represents the global optimization objective, measuring the average loss across all clients. $\mathcal{L}_k(\theta) = \mathbb{E}_{(\mathbf{Z}, y) \sim \mathcal{P}_k}[\ell(\mathbf{Z}, y)]$ denotes the local expected risk for client $k$ with data distribution $\mathcal{P}_k$. $\ell(\cdot)$ is the loss function evaluating the discrepancy between the model prediction and the true label $(\mathbf{Z}, y)$, and the participation weight $w_k \geq 0$ satisfies $\sum_k w_k = 1$. The training process alternates between local computation and global synchronization. During each communication round $t$, clients receive the global model $\theta^t$ and perform local updates:

$$\theta_k^{t+1} \leftarrow \theta_k^t - \eta \nabla_\theta \mathcal{L}_k(\theta_k^t), \tag{3}$$

where $\eta$ is the learning rate and $\nabla_\theta \mathcal{L}_k$ denotes the local gradient. The server then aggregates these updates through weighted averaging $\theta^{t+1} = \sum_{k=1}^{K} w_k \theta_k^{t+1}$.

## 4 METHODOLOGY

### 4.1 STRUCTURAL NOISE-AWARE AGGREGATION (SNAA)

**Motivation**. The effectiveness of FGL hinges on the ability to collaboratively train models across clients with diverse graph structures. However, structural noise in local graphs can significantly disrupt this process, introducing harmful inconsistencies in the learned representations. To address this challenge, we propose Structural Noise-Aware Aggregation (SNAA). Our method enables us to detect structurally corrupted clients and dynamically adjust their contributions to mitigate their adverse impact. Our approach is grounded in spectral graph theory, leveraging the observation in

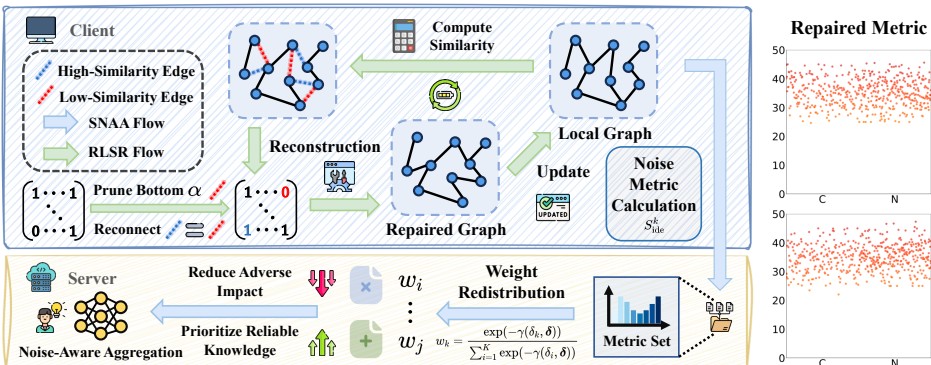

Figure 2: **Architecture illustration** of FedSDR. We used blue and green arrows to represent the two components of our method, SNAA and RLSR, respectively. In SNAA, each client computes the structural **noise metric** $S_{ide}^k$ locally, and the global model dynamically **reweights** their contributions based on this metric, thereby mitigating the influence of corrupted clients. In RLSR, we prune potentially corrupted edges with low feature similarity and reconnect an equivalent number of high-similarity edges to **align** the local graph structure with the global consensus. The right scatter plot displays fidelity metric distribution, showing that RLSR effectively **restores spectral properties**, which are critical for effective message-passing schemes.

Fig. 1 that structural noise manifests as deviations in the spectral properties of graphs. Details of SNAA are presented in Fig. 2.

**Structural Noise Detection**. The structural noise detection framework in SNAA stems from spectral graph analysis, where the Laplacian quadratic form $\mathbf{D}^T\mathbf{L}\mathbf{D}$ encodes critical structural properties of a graph, and its spectrum reflects the connectivity patterns and noise levels. Building on the spectral graph formula established in Eq. (1), we develop a rigorous structural noise evaluation metric. For client $k$ with Laplacian $\mathbf{L}^k = \mathbf{D}^k - \mathbf{A}^k$, the structural fidelity score $S_{ide}^k$ is computed locally to capture the extent of structural noise while preserving privacy:

$$S_{ide}^k = \frac{\langle \mathbf{D}^{k^T}\mathbf{L}^k\mathbf{D}^k, \mathbf{1}\rangle_F}{\langle \mathbf{D}^{k^T}\mathbf{D}^k, \mathbf{1}\rangle_F} = \frac{\sum_{i=1}^{N_k}\sum_{j=1}^{N_k}[\mathbf{D}^{k^T}\mathbf{L}^k\mathbf{D}^k]_{ij}}{\sum_{i=1}^{N_k}\sum_{j=1}^{N_k}[\mathbf{D}^{k^T}\mathbf{D}^k]_{ij}}, \tag{4}$$

where $\langle \cdot, \cdot \rangle_F$ denotes the Frobenius inner product and $\mathbf{1}$ is the all-ones matrix.

**Contribution Redistribution**. Building upon these noise assessments, we develop a principled contribution redistribution scheme for robust federated aggregation. The trustworthiness of each client $k$ is quantified through its structural fidelity metric $S_{ide}^k$, with lower values indicating worse preservation of authentic structural relationships. To mitigate noise-induced biases during model aggregation, we first compute the structural noise bias $\delta_k$ for each client, measuring its deviation from the global mean noise level while accounting for local node counts $N_k$:

$$\delta_k = S_{ide}^k - \frac{\sum_{i=1}^K N_i S_{ide}^i}{\sum_{j=1}^K N_j}. \tag{5}$$

This weighted formulation ensures fair comparison across clients with varying dataset sizes. A lower $\delta_k$ value indicates higher structural noise and thus lower reliability. The resulting bias terms are then normalized through min-max scaling to maintain consistent value ranges:

$$\gamma(\delta_k, \boldsymbol{\delta}) = -\frac{\delta_k \cdot \min(\boldsymbol{\delta})}{\max(\boldsymbol{\delta}) \cdot \min(\boldsymbol{\delta})}, \tag{6}$$

where $\boldsymbol{\delta}$ represents the complete set of client noise biases. The aggregation weight $w_k$ for client $k$ is obtained by applying the negative exponential function to the normalized bias, emphasizing clients with higher $S_{ide}^k$:

$$w_k = \frac{\exp(-\gamma(\delta_k, \boldsymbol{\delta}))}{\sum_{i=1}^K \exp(-\gamma(\delta_i, \boldsymbol{\delta}))}. \tag{7}$$

Through the exponential transformation, we assign higher weights to clients with cleaner structural patterns and smoothly alleviate the impact of noisier clients. Therefore, this weighting scheme ensures

that the global model prioritizes reliable knowledge from uncorrupted graphs, which is essential for maintaining consistent message-passing dynamics across clients.

**Reliability-Aware Weighting Aggregation**. We design a localized training mechanism to integrate local insights and global knowledge harmoniously. Each client iteratively refines its model using private data, preserving domain-specific expertise while mitigating catastrophic forgetting. The local training procedure generating $\theta_k^{t+1}$ ensures compatibility with our noise-aware aggregation. Given node features $\mathbf{X}^k \in \mathbb{R}^{N_k \times f}$ with $N_k$ nodes and $f$-dimensional features, $\mathcal{E}_k$ the set of edges in the graph of the client $k$, the forward pass yields node embeddings and logits:

$$\mathbf{Z}^k = f_{\theta_k}(\mathbf{X}^k, \mathcal{E}_k), \tag{8}$$

where $\mathbf{Z}^k \in \mathbb{R}^{N_k \times C}$ denotes the class logits for $C$ classes. Here, $f_{\theta_k}(\cdot)$ represents the local model inference. Let $\theta^t$ be the global model parameters at round $t$, and $\theta_k^t$ denote the local parameters for client $k$. For each node $v$ in the training set, $\mathbf{Z}_v^k \in \mathbb{R}^C$ represents the logits and $y_v^k \in \{1, ..., C\}$ is the true label. The training loss combines cross-entropy, label smoothing and model regularization:

$$\mathcal{L}(\theta_k^t) = \mathcal{L}_{ce} + \mathcal{L}_{ls} + \lambda \mathcal{L}_{reg} = \frac{1}{|\mathcal{M}_{\text{train}}|} \sum_{v \in \mathcal{M}_{\text{train}}} \sum_{y_v^k=1}^{C} \ell(\mathbf{Z}_v^k, y_v^k) + \frac{\epsilon}{2}\|\mathbf{Z}\|_2^2 + \lambda\|\theta_k^t - \theta^t\|_2^2, \tag{9}$$

where $\mathcal{M}_{\text{train}}$ is the training node mask, and $\lambda$ controls the regularization strength. The cross-entropy term $\ell(\cdot)$ follows the precise implementation:

$$\ell(\mathbf{Z}, y) = -\log\left(\frac{\exp(\mathbf{Z}[y])}{\sum_{c=1}^{C} \exp(\mathbf{Z}[c])}\right), \tag{10}$$

where $\epsilon$ controls label smoothing intensity. $\mathbf{Z}[y]$ and $\mathbf{Z}[c]$ index are the $y$-th and $c$-th elements of $\mathbf{Z}$, respectively. To guarantee differential privacy, we employ a Gaussian mechanism. The update rule for client $k$ is given by:

$$\theta_k^{t+1} \leftarrow \theta_k^t - \eta \nabla_{\theta_k^t} \mathcal{L}(\theta_k^t) + \mathcal{N}\left(0, \left(\frac{C\sigma}{|\mathcal{M}_{\text{train}}|}\right)^2 \mathbf{I}\right). \tag{11}$$

Here, $\eta$ is the learning rate, $C$ is the gradient clipping threshold, $\sigma$ is the noise multiplier. At the round $t + 1$, utilizing the aggregation weight $w_k$ defined in Eq. (7) for each client, the global model aggregates updates with reliability-aware weighting:

$$\theta^{t+1} = \sum_{k=1}^{K} w_k \theta_k^{t+1}. \tag{12}$$

By dynamically adjusting contributions based on spectral structural noise evaluation metrics, we align the global model with uncorrupted graph characteristics. Subsequently, we achieve robust training while maintaining model quality, efficiency, and privacy guarantees critical for practical deployment.

## 4.2 ROBUST LOCAL STRUCTURE RECONSTRUCTION (RLSR)

**Motivation**. While SNAA mitigates the global impact of structural noise, global model performance on corrupted clients remains compromised. Since directly reducing corrupted client contributions sacrifices their potentially valuable knowledge, it induces a fundamental mismatch in message-passing dynamics, thereby undermining global model performance on them. Our solution leverages global robust feature representations to guide local graph repair, simultaneously preserving authentic local patterns and correcting structural noise. Details of RLSR are presented in Fig. 2.

Since SNAA ensures the global model predominantly reflects uncorrupted structural patterns, its node embeddings encode discriminative feature relationships that are resilient to local noise. The core insight of RLSR lies in that structurally corrupted edges manifest as deviations in feature similarity space when evaluated against the global model. The normalized feature similarity matrix $\mathbf{C}^k \in \mathbb{R}^{N_k \times N_k}$ captures pairwise relationships:

$$\mathbf{C}^k = \text{diag}(\mathbf{H}^k \mathbf{H}^{k^T})^{-\frac{1}{2}} (\mathbf{H}^k \mathbf{H}^{k^T}) \text{diag}(\mathbf{H}^k \mathbf{H}^{k^T})^{-\frac{1}{2}}, \tag{13}$$

where $\mathbf{H}^k = f_\theta(\mathbf{X}^k, \mathcal{E}_k) \in \mathbb{R}^{N_k \times d}$ be node embeddings from the global model $f_\theta$. Each element in Eq. (13) measures the feature similarity between nodes $u$ and $v$ under the global inductive inference:

$$\mathbf{C}_{uv}^k = \frac{\langle \mathbf{h}_u^k, \mathbf{h}_v^k \rangle}{\|\mathbf{h}_u^k\| \cdot \|\mathbf{h}_v^k\|}, \tag{14}$$

where $\mathbf{h}_u^k$ denotes the embedding of node $u$ in client $k$. Authentic edges in uncorrupted graphs align with high feature similarities, whereas structural noise introduces edges with discordantly low similarities. RLSR exploits this discrepancy through two edge refinement processes.

First, to fundamentally eliminate structural noise, we prune corrupted edges. For each client $k$, we compute a dynamic pruning threshold $\tau_p^k$ as the $\alpha$-quantile of existing edge similarities, defined formally through the quantile function $Q(\cdot, \alpha)$:

$$\tau_p^k = Q\left(\{\mathbf{C}_{uv}^k \mid \mathbf{A}_{uv}^k = 1\}, \alpha\right), \tag{15}$$

where $Q(S, \alpha)$ denotes that there are $\alpha$ proportion of elements in $S$ whose values are less than or equal to it, i.e., $\alpha$ proportion of connected edges have similarities below $\tau_p^k$. Edges satisfying $\mathbf{C}_{uv}^k < \tau_p^k$ are identified as corrupted connections and removed. This operation eliminates connections that contradict the healthy knowledge of the global model, preserving only those consistent with the consensus of global features.

Second, we conduct feature-aware edge reconnection. Merely removing noisy edges risks over-sparsification, which disrupts message-passing dynamics. Subsequent reconnection compensates for pruned edges by establishing high-similarity connections absent in the original structure. Candidate edges are sampled from non-adjacent node pairs with similarities exceeding a reconnection threshold $\tau_r^k$, defined as:

$$\tau_r^k = Q\left(\{\mathbf{C}_{uv}^k \mid \mathbf{A}_{uv}^k = 0, u \neq v\}, 1 - \frac{2\alpha|\mathcal{E}^k|}{N_k(N_k - 1) - 2|\mathcal{E}^k|}\right), \tag{16}$$

where $\tau_r^k$ identifies the top $\alpha|\mathcal{E}^k|$ highest-similarity non-edge, matching the count of pruned edges. New edges are established for pairs $(u, v)$ satisfying $\mathbf{C}_{uv}^k \geq \tau_r^k$, while excluding self-loop reconnection to improve the message-passing scheme. Therefore, reconnected edges exhibit strong feature alignment with the global consensus. The joint pruning-reconnection operation in Eq. (15) and Eq. (16) yields a repaired adjacency matrix $\tilde{\mathbf{A}}^k$:

$$\tilde{\mathbf{A}}_{uv}^k = \begin{cases} 0 & \text{if } \mathbf{A}_{uv}^k = 1 \text{ and } \mathbf{C}_{uv}^k < \tau_p^k, \\ 1 & \text{if } \mathbf{A}_{uv}^k = 0 \text{ and } u \neq v \text{ and } \mathbf{C}_{uv}^k \geq \tau_r^k, \\ \mathbf{A}_{uv}^k & \text{otherwise.} \end{cases} \tag{17}$$

By replacing corrupted edges with feature-consistent connections, we succeed in both mitigating structural noise and preserving client-specific patterns unaffected by corruption. In this way, RLSR fundamentally addresses issues of structural noise, establishing a robust foundation for decentralized GNN training in high-structural-noise environments.

## 5 EXPERIMENTS

### 5.1 EXPERIMENTAL SETUP

**Datasets.** We conducted experiments under various structural noise scenarios to validate the superiority of our proposed FedSDR. The datasets with different scales include homophilic datasets PubMed Sen et al. (2008), Coauthor-CS, Coauthor-Physics Shchur et al. (2018a), ogbn-products Hu et al. (2020) and heterophilic datasets Actor Pei et al. (2020), Roman-empire Baranovskiy et al. (2023), ogbn-mag Hu et al. (2020). Detailed descriptions of these datasets can be found in Appendix B.

**Baselines.** We compare FedSDR with several state-of-the-art federated approaches. (1) **FedAvg** McMahan et al. (2017a); (2) **FedGTA** Li et al. (2024d); (3) **FedProto** Tan et al. (2022); (4) **FedProx** Li et al. (2020b); (5) **FedTAD** Zhu et al. (2024) applying topology-aware knowledge distillation technology; (6) **FGSSL** Huang et al. (2023) which reveals distortion brought by node-level semantics and graph-level structure, (7) **MOON** Li et al. (2021a), (8) **Scaffold** Karimireddy et al. (2020), (9) **Ditto** Li et al. (2021b) and (10) **RHFL** Fang & Ye (2022a), two state-of-the-art robust FL methods; (11) **FedSage+** Zhang et al. (2021a) and (12) **FED-PUB** Baek et al. (2023), two state-of-the-art subgraph FL baselines; (13) **AdaFGL** Li et al. (2024c) tackling topology heterogeneity, (14) **FedSSP** Tan et al. (2024) accommodating personalized preference.

**Implementation Details.** We use the Louvain algorithm Blondel et al. (2008) to partition the graph, which is an effective strategy to obtain non-IID data in FGL Huang et al. (2023); Zhang et al. (2021b).

For homophilic datasets, we adopt a 20%/40%/40% split for training, validation, and testing, whereas for heterophilic datasets, the split is 50%/25%/25%. The local GNN models are trained using the Adam optimizer Kinga et al. (2015). The specific learning rate, the number of communication rounds and clients are detailed in Appendix B. To simulate high-structural-noise scenarios, the corruption ratio (proportion of corrupted clients), the noise extent (proportion of edges randomly added and then deleted in corrupted clients), and the pruning proportion $\alpha$ are set to 1, 0.5, and 0.3, respectively. The results represent the average of 5 runs with different random seeds.

## 5.2 EXPERIMENTAL RESULTS

**Performance Comparison** We present federated graph classification results across seven datasets, with the final average test accuracy shown in Tab. 1. The results demonstrate that FedSDR markedly outperforms all baseline methods. Structure-aware approaches like FGSSL and FedGTA exhibit more stable performance across datasets. While MOON achieves competitive results on some datasets by aligning local representations with the global consensus, it suffers significant performance degradation on others due to its inability to repair corrupted graph structure. Notably, traditional FL algorithms such as FedAvg and FedProx surpass several more complex methods, highlighting how severe structural noise substantially impacts most existing approaches.

| Methods | PubMed | Coauthor-CS | Coauthor-Phy | Actor | Roman-empire | ogbn-mag | ogbn-products |
|---|---|---|---|---|---|---|---|
| FedAvg [ASTAT17] | $78.41 \pm 0.36$ | $82.01 \pm 0.49$ | $86.37 \pm 0.27$ | $31.28 \pm 0.77$ | $41.72 \pm 0.39$ | $42.74 \pm 0.22$ | $72.66 \pm 1.53$ |
| FedProx [arxiv18] | $77.82 \pm 0.97$ | $74.94 \pm 0.36$ | $86.19 \pm 0.20$ | $31.27 \pm 0.50$ | $41.76 \pm 0.71$ | $39.93 \pm 0.49$ | $70.51 \pm 0.92$ |
| Scaffold [ICML20] | $41.76 \pm 0.83$ | $43.00 \pm 3.23$ | $77.41 \pm 1.75$ | $23.52 \pm 1.35$ | $18.08 \pm 0.33$ | $15.09 \pm 3.40$ | $59.07 \pm 2.21$ |
| MOON [CVPR21] | $68.55 \pm 2.27$ | $75.91 \pm 0.29$ | $88.14 \pm 0.16$ | $31.14 \pm 0.55$ | $42.51 \pm 0.30$ | $42.96 \pm 0.84$ | $67.94 \pm 0.45$ |
| Ditto [ICML21] | $76.84 \pm 1.02$ | $80.33 \pm 0.77$ | $85.49 \pm 0.55$ | $30.62 \pm 0.29$ | $41.30 \pm 0.67$ | $41.23 \pm 1.20$ | $71.89 \pm 0.95$ |
| FedSage+ [NIPS21] | $78.29 \pm 0.93$ | $80.72 \pm 0.51$ | $85.83 \pm 1.28$ | $30.42 \pm 0.74$ | $41.64 \pm 0.57$ | $42.92 \pm 0.47$ | $71.98 \pm 0.40$ |
| FedProto [AAAI22] | $75.24 \pm 0.17$ | $67.26 \pm 0.40$ | $84.42 \pm 0.23$ | $22.38 \pm 0.33$ | $19.42 \pm 0.18$ | $37.16 \pm 0.29$ | $66.79 \pm 1.59$ |
| RHFL [CVPR22] | $78.07 \pm 0.77$ | $79.76 \pm 0.87$ | $86.11 \pm 1.05$ | $31.27 \pm 0.66$ | $42.33 \pm 0.45$ | $42.80 \pm 0.41$ | $71.04 \pm 0.58$ |
| FGSSL [IJCAI23] | $77.65 \pm 0.41$ | $81.05 \pm 0.63$ | $84.78 \pm 1.61$ | $31.20 \pm 0.64$ | $37.41 \pm 0.51$ | $41.23 \pm 0.75$ | $69.63 \pm 1.13$ |
| FED-PUB [ICML23] | $75.28 \pm 0.64$ | $79.34 \pm 0.45$ | $85.55 \pm 0.87$ | $30.59 \pm 0.28$ | $42.12 \pm 0.65$ | $40.15 \pm 0.56$ | $70.06 \pm 0.99$ |
| FedTAD [IJCAI24] | $77.92 \pm 1.02$ | $68.90 \pm 0.93$ | OOM | $31.13 \pm 0.74$ | $41.29 \pm 0.40$ | OOM | OOM |
| AdaFGL [ICDE24] | $76.63 \pm 0.88$ | $79.50 \pm 0.56$ | $87.19 \pm 0.44$ | $30.90 \pm 0.62$ | $41.82 \pm 0.57$ | $43.22 \pm 0.34$ | $69.68 \pm 1.05$ |
| FedGTA [VLDB24] | $78.04 \pm 0.58$ | $81.24 \pm 0.61$ | $85.77 \pm 0.16$ | $31.03 \pm 0.93$ | $40.65 \pm 0.21$ | $42.52 \pm 0.33$ | $71.52 \pm 0.78$ |
| FedSSP [NIPS24] | $77.46 \pm 0.43$ | $79.67 \pm 0.23$ | $86.12 \pm 0.74$ | $30.52 \pm 0.70$ | $39.45 \pm 0.62$ | $40.48 \pm 0.63$ | $69.36 \pm 0.85$ |
| **FedSDR (ours)** | $\mathbf{82.57 \pm 0.64}$ | $\mathbf{82.29 \pm 0.32}$ | $\mathbf{89.37 \pm 0.24}$ | $\mathbf{32.36 \pm 0.59}$ | $\mathbf{48.33 \pm 0.28}$ | $\mathbf{47.41 \pm 0.42}$ | $\mathbf{78.57 \pm 0.28}$ |

Table 1: Comparison with the state-of-the-art methods on non-IID data. The best and second results are highlighted with bold and underline, respectively. Please see additional results in Appendix A.

**Convergence Analysis** Fig. 3 illustrates the training curves of the average test accuracy with standard deviation across five random runs of three datasets (Actor, PubMed, Roman-empire), comparing FedSDR with various baseline methods. The results demonstrate that existing approaches exhibit significant performance degradation in high-structural-noise scenarios. In contrast, our proposed FedSDR maintains both stable convergence and superior performance, showcasing its robustness to structural noise.

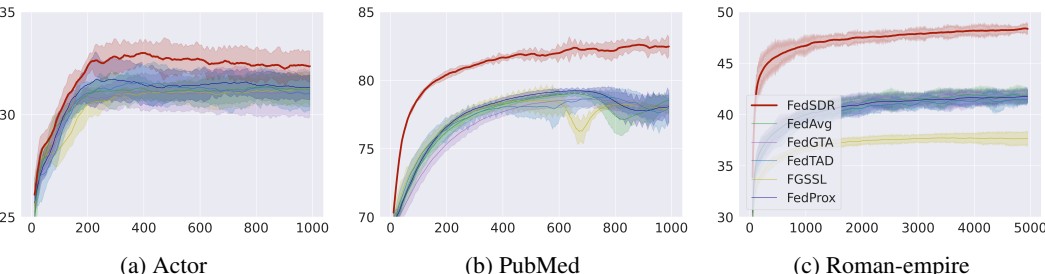

(a) Actor        (b) PubMed        (c) Roman-empire

Figure 3: Test accuracy curves of FedSDR and five other methods on three datasets (Actor, PubMed, Roman-empire), with accuracy (%) on the y-axis and communication rounds on the x-axis.

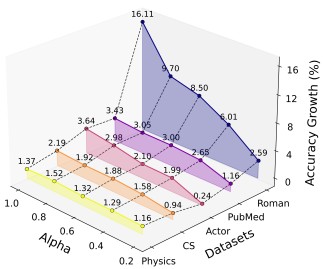

Figure 4: Varying Corruption.    Table 2: **Ablation study** of FedSDR.    Figure 5: Parameter $\alpha$.

| SNAA | RLSR | Datasets | | |
|---|---|---|---|---|
| | | PubMed | Actor | Physics |
| ✗ | ✗ | 78.41 | 31.28 | 86.37 |
| ✓ | ✗ | 81.82 | 31.56 | 88.92 |
| ✗ | ✓ | 82.26 | 32.07 | 89.21 |
| ✓ | ✓ | **82.57** | **32.36** | **89.37** |

### 5.3 Varying Corruption Ratios and Noise Extents

We evaluate the robustness of FedSDR under varying degrees of structural corruption on PubMed. As demonstrated in Fig. 4, FedSDR achieves consistent performance improvements over FedAvg across five representative structural corruption ratios and noise extents, accounting for all potential cases. Our analysis reveals two critical findings: (1) While FedAvg exhibits progressively severe performance degradation with increasing noise extent, FedSDR maintains stable accuracy; (2) Even under high corruption ratios, our adaptive aggregation and local graph reconstruction enable robust collaboration, consistently outperforming the baseline. These results validate the effectiveness of FedSDR in handling diverse structural corruption scenarios.

### 5.4 Ablation Study

To comprehensively evaluate the contribution of each component in FedSDR, we conducted an ablation study across three datasets. Tab. 2 reveals that SNAA alone improves accuracy over FedAvg through effectively identifying and mitigating structural noise during global aggregation. Furthermore, RLSR achieves noticeable success in reconstructing local corrupted graph structures and restoring message-passing dynamics. The complete FedSDR framework, combining both components, delivers optimal performance by addressing structural noise at both global and local levels. This analysis confirms that both global noise-aware aggregation and local structural reconstruction are indispensable for effective FGL training under structural corruption.

### 5.5 Hyper-parameter Study

We conduct a thorough investigation of the pruning proportion parameter $\alpha$ to understand its impact on model performance and robustness. As shown in Fig. 5, FedSDR outperforms FedAvg across all tested $\alpha$ values (ranging from 0.2 to 1.0) on five datasets. Moreover, continuous performance improvement with increasing values of $\alpha$ indicates that our reconstruction successfully distinguishes between corrupted and meaningful edges, enabling effective noise removal while preserving valuable structural relationships. This performance advantage across parameter configurations demonstrates the robustness of our approach.

## 6 Conclusion

This paper first presents a comprehensive study addressing structural noise. We propose **FedSDR**, a novel framework that combines spectral-guided noise detection and adaptive structural repair. For global collaboration, **SNAA** evaluates and mitigates structural corruption by dynamically reweighting client contributions during global collaboration. For local deployment, **RLSR** reconstructs corrupted graphs by aligning them with the global consensus while preserving valid local connectivity. Extensive experiments across multiple datasets and noise scenarios demonstrate that FedSDR consistently outperforms state-of-the-art methods, establishing a trustworthy paradigm for FGL in high-structural-noise environments.

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

## A  ADDITIONAL EXPERIMENTS

We comprehensively compare FedSDR with state-of-the-art methods under varying **corruption ratios** (the proportion of structurally corrupted clients) and **noise extents** (the degree of structural corruption per corrupted client). Our experimental setup evaluates three corruption ratios of 0.3, 0.5, and 1 (the extreme case detailed in Tab. 1), with the noise extent and pruning proportion $\alpha$ fixed at 0.5 and 0.3. In addition, we validate the robustness of FedSDR across three noise extents (0.3, 0.5 shown in Tab. 4, and 1), while keeping the corruption ratio (0.5) and pruning proportion $\alpha$ (0.3) constant. Notably, as the corruption severity increases, competing methods exhibit significant performance degradation, whereas FedSDR maintains stable performance, demonstrating a substantial advantage. The best and second results are highlighted with bold and underline, respectively.

| Methods | PubMed | Coauthor-CS | Coauthor-Phy | Actor | Roman-empire | ogbn-mag | ogbn-products |
|---|---|---|---|---|---|---|---|
| FedAvg [ASTAT17] | $81.83 \pm 0.69$ | $81.69 \pm 0.47$ | $89.08 \pm 0.27$ | $31.37 \pm 0.58$ | $39.23 \pm 0.50$ | $40.34 \pm 0.57$ | $73.34 \pm 0.49$ |
| FedProx [arxiv18] | $82.23 \pm 0.68$ | $82.05 \pm 0.22$ | $89.50 \pm 0.29$ | $\underline{31.57 \pm 0.65}$ | $39.74 \pm 0.54$ | $39.92 \pm 0.46$ | $73.06 \pm 0.47$ |
| Scaffold [ICML20] | $40.45 \pm 1.81$ | $48.57 \pm 2.43$ | $79.88 \pm 0.44$ | $23.69 \pm 0.46$ | $17.90 \pm 0.32$ | $16.77 \pm 0.42$ | $66.28 \pm 0.41$ |
| MOON [CVPR21] | $76.65 \pm 2.76$ | $\underline{83.36 \pm 0.18}$ | $\mathbf{91.06 \pm 0.16}$ | $31.36 \pm 0.69$ | $\underline{39.97 \pm 0.17}$ | $40.33 \pm 0.37$ | $73.47 \pm 0.36$ |
| Ditto [ICML21] | $78.92 \pm 0.89$ | $81.28 \pm 0.75$ | $88.61 \pm 0.49$ | $30.71 \pm 0.27$ | $38.26 \pm 0.33$ | $41.08 \pm 0.79$ | $71.95 \pm 0.82$ |
| FedSage+ [NIPS21] | $81.73 \pm 0.68$ | $81.73 \pm 0.23$ | $89.11 \pm 1.02$ | $30.95 \pm 0.52$ | $39.61 \pm 0.69$ | $\underline{42.60 \pm 0.41}$ | $72.85 \pm 0.40$ |
| FedProto [AAAI22] | $82.01 \pm 0.40$ | $74.14 \pm 0.29$ | $86.70 \pm 0.23$ | $23.45 \pm 0.29$ | $13.93 \pm 0.48$ | $38.03 \pm 0.56$ | $69.20 \pm 0.39$ |
| RHFL [CVPR22] | $82.03 \pm 0.53$ | $81.45 \pm 0.66$ | $89.08 \pm 0.41$ | $31.06 \pm 0.26$ | $39.88 \pm 0.45$ | $42.04 \pm 0.62$ | $\underline{74.58 \pm 0.57}$ |
| FGSSL [IJCAI23] | $82.09 \pm 0.41$ | $80.97 \pm 0.57$ | $89.94 \pm 0.33$ | $30.97 \pm 0.52$ | $33.16 \pm 0.32$ | $41.26 \pm 0.63$ | $72.32 \pm 0.24$ |
| FED-PUB [ICML23] | $80.27 \pm 0.54$ | $80.84 \pm 0.56$ | $87.90 \pm 0.34$ | $30.87 \pm 0.73$ | $38.39 \pm 0.47$ | $40.45 \pm 0.29$ | $71.53 \pm 0.27$ |
| FedTAD [IJCAI24] | $\underline{82.62 \pm 0.43}$ | $73.16 \pm 1.55$ | OOM | $30.79 \pm 0.45$ | $39.83 \pm 0.80$ | OOM | OOM |
| AdaFGL [ICDE24] | $81.32 \pm 0.71$ | $80.47 \pm 0.37$ | $88.75 \pm 0.48$ | $31.21 \pm 0.59$ | $39.28 \pm 0.50$ | $42.36 \pm 0.78$ | $72.94 \pm 0.66$ |
| FedGTA [VLDB24] | $82.20 \pm 0.75$ | $81.03 \pm 0.60$ | $85.77 \pm 0.16$ | $31.30 \pm 0.50$ | $38.80 \pm 1.12$ | $42.33 \pm 0.69$ | $73.78 \pm 0.53$ |
| FedSSP [NIPS24] | $79.90 \pm 0.32$ | $79.22 \pm 0.39$ | $88.04 \pm 0.50$ | $30.74 \pm 0.81$ | $37.52 \pm 0.65$ | $40.28 \pm 0.47$ | $72.02 \pm 0.35$ |
| **FedSDR (ours)** | $\mathbf{84.07 \pm 0.48}$ | $\mathbf{83.43 \pm 0.39}$ | $\underline{90.72 \pm 0.43}$ | $\mathbf{33.30 \pm 0.42}$ | $\mathbf{41.58 \pm 0.38}$ | $\mathbf{46.88 \pm 0.21}$ | $\mathbf{80.57 \pm 0.42}$ |

Table 3: Corruption Ratio = 0.3, Noise Extent = 0.5, $\alpha = 0.3$

| Methods | PubMed | Coauthor-CS | Coauthor-Phy | Actor | Roman-empire | ogbn-mag | ogbn-products |
|---|---|---|---|---|---|---|---|
| FedAvg [ASTAT17] | $81.28 \pm 2.02$ | $81.26 \pm 0.26$ | $88.20 \pm 0.22$ | $31.29 \pm 0.53$ | $39.14 \pm 0.60$ | $43.56 \pm 0.81$ | $72.31 \pm 0.24$ |
| FedProx [arxiv18] | $80.82 \pm 0.96$ | $\underline{82.48 \pm 0.51}$ | $88.68 \pm 0.13$ | $\underline{31.56 \pm 0.59}$ | $39.26 \pm 0.50$ | $42.48 \pm 0.27$ | $71.83 \pm 0.87$ |
| Scaffold [ICML20] | $40.71 \pm 0.90$ | $44.72 \pm 4.54$ | $80.21 \pm 1.61$ | $23.06 \pm 1.37$ | $16.76 \pm 1.86$ | $17.29 \pm 2.02$ | $59.90 \pm 2.03$ |
| MOON [CVPR21] | $77.09 \pm 2.59$ | $80.92 \pm 0.54$ | $\underline{90.08 \pm 0.17}$ | $31.04 \pm 0.64$ | $40.41 \pm 0.22$ | $\underline{44.99 \pm 0.52}$ | $72.07 \pm 0.22$ |
| Ditto [ICML21] | $77.95 \pm 0.85$ | $80.31 \pm 0.82$ | $86.91 \pm 0.32$ | $30.03 \pm 0.28$ | $38.32 \pm 0.24$ | $42.85 \pm 1.01$ | $71.27 \pm 0.25$ |
| FedSage+ [NIPS21] | $79.91 \pm 0.82$ | $81.38 \pm 0.57$ | $88.90 \pm 0.81$ | $31.42 \pm 0.43$ | $41.64 \pm 0.57$ | $42.92 \pm 0.47$ | $\underline{72.86 \pm 0.38}$ |
| FedProto [AAAI22] | $79.95 \pm 0.58$ | $72.46 \pm 0.43$ | $85.55 \pm 0.21$ | $23.50 \pm 0.28$ | $15.94 \pm 0.13$ | $37.27 \pm 0.89$ | $70.13 \pm 0.44$ |
| RHFL [CVPR22] | $78.86 \pm 0.67$ | $82.07 \pm 0.49$ | $87.34 \pm 0.35$ | $31.40 \pm 0.48$ | $42.33 \pm 0.45$ | $42.31 \pm 0.92$ | $\underline{73.25 \pm 0.29}$ |
| FGSSL [IJCAI23] | $77.65 \pm 0.41$ | $81.05 \pm 0.63$ | $88.60 \pm 0.76$ | $31.31 \pm 0.30$ | $34.58 \pm 0.56$ | $40.38 \pm 0.30$ | $72.10 \pm 0.34$ |
| FED-PUB [ICML23] | $75.82 \pm 0.59$ | $79.79 \pm 0.37$ | $85.28 \pm 0.31$ | $30.77 \pm 0.35$ | $42.12 \pm 0.65$ | $40.15 \pm 0.56$ | $71.32 \pm 0.46$ |
| FedTAD [IJCAI24] | $80.93 \pm 0.58$ | $74.81 \pm 1.38$ | OOM | $31.25 \pm 0.57$ | $\underline{40.53 \pm 0.46}$ | OOM | OOM |
| AdaFGL [ICDE24] | $78.74 \pm 0.60$ | $80.06 \pm 0.74$ | $87.90 \pm 0.65$ | $31.06 \pm 0.32$ | $40.36 \pm 0.63$ | $44.67 \pm 0.30$ | $71.24 \pm 0.65$ |
| FedGTA [VLDB24] | $\underline{81.30 \pm 0.77}$ | $81.22 \pm 0.49$ | $88.28 \pm 0.18$ | $31.25 \pm 0.22$ | $39.64 \pm 0.76$ | $42.12 \pm 0.35$ | $72.01 \pm 0.63$ |
| FedSSP [NIPS24] | $78.24 \pm 0.45$ | $78.88 \pm 0.71$ | $87.78 \pm 0.92$ | $30.58 \pm 0.73$ | $41.45 \pm 0.62$ | $40.97 \pm 0.58$ | $71.81 \pm 0.32$ |
| **FedSDR (ours)** | $\mathbf{84.06 \pm 0.44}$ | $\mathbf{83.10 \pm 0.32}$ | $\mathbf{90.50 \pm 0.23}$ | $\mathbf{33.10 \pm 0.50}$ | $\mathbf{45.12 \pm 0.36}$ | $\mathbf{48.06 \pm 0.20}$ | $\mathbf{80.23 \pm 1.20}$ |

Table 4: Corruption Ratio = 0.5, Noise Extent = 0.3, $\alpha = 0.3$

## B    DATASETS

The statistics and configuration of the datasets used in our experiments are provided in Tab. 5.

- **PubMed:** PubMed Sen et al. (2008) is a widely adopted citation network dataset in which nodes represent academic papers and edges correspond to citation relationships between them. Each node is associated with a word vector feature encoding the presence or absence of specific keywords in the corresponding paper. As a standard benchmark in graph-based machine learning, this dataset is commonly employed for node classification tasks, particularly in FL scenarios where data privacy must be preserved.

- **Coauthor-CS, Coauthor-Physics:** Derived from the Microsoft Academic Graph, the Coauthor-CS and Coauthor-Physics datasets Shchur et al. (2018a) originate from the KDD Cup 2016 challenge. These benchmark datasets represent academic collaboration networks, with nodes corresponding to authors and edges indicating co-authorship. Coauthor-CS comprises 18,333 nodes and 81,894 edges, featuring node attributes based on paper keywords and 15 distinct research fields as class labels. The larger Coauthor-Physics dataset contains 34,493 nodes and 247,962 edges, with similar node features and labels representing classification into 5 main research areas. Both datasets serve as established benchmarks for evaluating graph neural networks, particularly for node classification tasks, due to their realistic academic network structures and comprehensive feature representations.

- **Actor:** The Actor dataset Pei et al. (2020) is an actor co-occurrence network constructed from Wikipedia, where nodes represent actors and edges indicate co-appearance in Wikipedia pages. Each node is characterized by a bag-of-words feature vector derived from the corresponding Wikipedia page content. The dataset includes five actor categories, determined through semantic analysis of their associated Wikipedia entries. As a standard benchmark in graph machine learning, it is widely used to evaluate node classification tasks and related graph-based learning problems.

- **Roman-empire:** The Roman-empire dataset Baranovskiy et al. (2023) is derived from the English Wikipedia article about the Roman Empire. Nodes represent word occurrences (including non-unique words), with the graph size reflecting the article length. Edges are established based on two linguistic relationships: (1) sequential co-occurrence when words appear consecutively in the text, and (2) syntactic dependencies from the sentence parse trees. This construction yields a chain-like graph augmented with linguistic connections, capturing both sequential and syntactic relationships between words.

- **ogbn-mag** The ogbn-mag dataset from the Open Graph Benchmark (OGB) facilitates node property prediction within a heterogeneous academic network derived from the Microsoft Academic Graph (MAG). This graph comprises four entity types—papers (736,389 nodes), authors (1.13 million nodes), institutions (8,740 nodes), and fields of study (59,965 nodes)—interconnected by directed relations such as authorship, citation, and affiliation. Each paper node possesses a 128-dimensional word2vec feature vector, while other entities lack initial features. The objective is a 349-class classification to predict the publishing venue of each paper. A temporally realistic split is employed, where papers published before 2018 are used for training, those from 2018 for validation, and papers since 2019 for testing, ensuring a predictive task that forecasts future trends based on historical data Hu et al. (2020).

- **ogbn-products** The ogbn-products dataset from the Open Graph Benchmark (OGB) supports node property prediction tasks within an Amazon product co-purchasing network. This undirected unweighted graph includes approximately 2.4 million nodes, each representing a product, and 61.9 million edges indicating frequent co-purchase relationships. Each node has a 100-dimensional feature vector derived from product descriptions via bag-of-words and PCA. The task is to predict the product category among 47 top-level classes in a multi-class classification setup. The dataset is split by sales rank, with the top 8% of products for training, the next 2% for validation, and the remaining 90% for testing, simulating a realistic scenario where popular products have labeled data while predictions are required for less popular items Hu et al. (2020).

## C    ETHICS STATEMENT

This work adheres to the ICLR Code of Ethics. In this study, no human subjects or animal experimentation was involved. All datasets used were sourced in compliance with relevant usage guidelines,

| Dataset | Nodes | Edges | Classes | Features | Learning Rate | Rounds | Clients |
|---------|-------|-------|---------|----------|---------------|--------|---------|
| PubMed | 19,717 | 44,338 | 3 | 500 | 0.01 | 1,000 | 10 |
| Coauthor-CS | 18,333 | 81,894 | 15 | 6,805 | 0.005 | 1,000 | 50 |
| Coauthor-Phy | 34,493 | 247,962 | 5 | 8,415 | 0.01 | 1,000 | 100 |
| Actor | 7,600 | 29,926 | 5 | 931 | 0.002 | 1,000 | 20 |
| Roman-empire | 22,662 | 32,927 | 18 | 300 | 0.02 | 5,000 | 50 |
| ogbn-mag | 1,939,743 | | 349 | 128 | 0.01 | 1,000 | 100 |
| ogbn-products | 2,449,029 | 61,859,140 | 47 | 100 | 0.01 | 1,000 | 100 |

Table 5: **Statistics** of datasets used in experiments.

ensuring no violation of privacy. We have taken care to avoid any biases or discriminatory outcomes in our research process. No personally identifiable information was used, and no experiments were conducted that could raise privacy or security concerns. We are committed to maintaining transparency and integrity throughout the research process.

## D    REPRODUCIBILITY STATEMENT

We have made every effort to ensure that the results presented in this paper are reproducible. All code and datasets have been made publicly available in an anonymous repository to facilitate replication and verification. The experimental setup, including training steps, model configurations, and hardware details, is described in detail in the paper. We have also provided a full description of FedSDR, to assist others in reproducing our experiments.

Additionally, all datasets are publicly available, ensuring consistent and reproducible evaluation results.

We believe these measures will enable other researchers to reproduce our work and further advance the field.

## E    LLM USAGE

Large Language Models (LLMs) were used to aid in the writing and polishing of the manuscript. Specifically, we used an LLM to assist in refining the language, improving readability, and ensuring clarity in various sections of the paper. The model helped with tasks such as sentence rephrasing, grammar checking, and enhancing the overall flow of the text.

It is important to note that the LLM was not involved in the ideation, research methodology, or experimental design. All research concepts, ideas, and analyses were developed and conducted by the authors. The contributions of the LLM were solely focused on improving the linguistic quality of the paper, with no involvement in the scientific content or data analysis.

The authors take full responsibility for the content of the manuscript, including any text generated or polished by the LLM. We have ensured that the LLM-generated text adheres to ethical guidelines and does not contribute to plagiarism or scientific misconduct.

