# OpenReview forum: "FedSDR: Federated Graph Learning with Structural Noise Detection and Reconstruction"
_ICLR.cc/2026/Conference — ICLR 2026 Conference Withdrawn Submission_

### Official Review · Reviewer_gFac · 2025-10-27

**Soundness:** 2
**Presentation:** 3
**Contribution:** 2
**Rating:** 2
**Confidence:** 5

**Summary:**

This paper introduces FedSDR, a robust server–client collaborative algorithm designed to mitigate the impact of low-quality topological perturbations in federated graph learning. Extensive empirical studies substantiate the effectiveness of the proposed approach.

**Strengths:**

1. The empirical analysis provides valuable insights into the research context and effectively reinforces the authors’ arguments.

2. The presentation of sections, figures, and tables is clear, coherent, and visually appealing.

3. The discussion of baseline methods is appropriate and demonstrates the authors’ solid understanding of related work.

**Weaknesses:**

1. The research topic—leveraging topological structures in federated graph learning—while important, lacks novelty. Similar themes have been extensively explored in prior studies (exists in nearly all federated graph learning studies). Moreover, topology reconstruction within federated settings has been widely discussed, which diminishes the originality and impact of this work.

2. The authors attempt to quantify topological quality using the Laplacian quadratic from spectral graph theory to inspire model framework design. While this approach is intuitively appealing, several key aspects remain insufficiently discussed. For instance, why should structural noise in federated graphs be inherently related to graph connectivity as reflected by the Laplacian quadratic? Under the broader paradigm of attributed graph learning, could node features and labels have a greater influence on topology than degree-based connectivity alone? I strongly encourage the authors to clarify the precise definition of structural noise in federated graph learning, as well as the fundamental motivation and intuition behind using the Laplacian quadratic in this context.

3. The paper’s integration of Laplacian quadratic–based coordination between server-side noise-aware aggregation and client-side topology augmentation is commendable. However, the core design ideas underlying these modules have already been explored in prior works such as weighted aggregation in FedSpray and data augmentation in FedSage+. As a result, the overall novelty of FedSDR is substantially weakened.

4. Although the experiments are relatively comprehensive, the study’s focus on topological issues in federated graph learning warrants a tighter alignment with relevant research in this specific domain. Comparisons with methods like FedProto are somewhat misplaced, as their objectives differ from this work’s focus. Instead, the paper should emphasize comparisons with more directly related approaches—such as FedSpray (KDD 2024), FedATH (ICML 2025), and FedIIH (AAAI 2025)—to better contextualize its contributions within the federated graph learning community.

**Questions:**

Please see the above weakness.

**Details Of Ethics Concerns:**

None.

---

### Official Review · Reviewer_BUvp · 2025-10-30

**Soundness:** 3
**Presentation:** 3
**Contribution:** 3
**Rating:** 6
**Confidence:** 3

**Summary:**

This paper proposes an innovative framework for addressing structural noise in federated graph learning, systematically explores the under-studied key issue of "structural noise" in federated graph learning, and clearly points out the limitations of existing methods at both the global and local levels.

**Strengths:**

A. The proposed FedSDR framework comprises two core components, SNAA and RLSR, addressing the problem from both global and local perspectives respectively. Its clear logic forms a closed loop. In particular, its approach of utilizing spectral theory for noise detection and graph inpainting based on global model feature similarity is theoretically sound and innovative.

B. The experimental section covers various datasets, including both identically matched and dissimilarly matched graphs. It compares the proposed method with numerous baseline methods and includes convergence analysis, ablation experiments, hyperparameter studies, and robustness tests under different noise scenarios. The results strongly support the effectiveness of the proposed method.

**Weaknesses:**

A. While mentioning the use of Gaussian mechanisms for differential privacy, the RLSR process requires the client to upload feature similarity information or receive global model embeddings for graph repair, which may introduce additional privacy risks. This is not analyzed or evaluated in depth in the paper.

B. SNAA requires calculating the Laplacian matrix of the graph, and RLSR requires calculating feature similarity across the entire graph and performing edge operations. Both of these increase the computational burden on the client and the number of communication rounds/data volume between the client and the server. The paper lacks a quantitative evaluation of the overall overhead of the FedSDR framework.

C. The paper lacks rigorous theoretical guarantees or analysis of the proposed method (especially the validity boundary of the spectral metric Side and the structural properties of the graph after RLSR repair).

**Questions:**

see weaknesses

---

### Official Review · Reviewer_w6Sb · 2025-11-01

**Soundness:** 3
**Presentation:** 3
**Contribution:** 3
**Rating:** 4
**Confidence:** 3

**Summary:**

This paper addresses a crucial challenge in Federated Graph Learning (FGL): the presence of structural noise in decentralized subgraphs, which can degrade the global model due to harmful message-passing conflicts. The authors introduce FedSDR, a robust FGL framework composed of two key components:
* SNAA (Structural Noise-Aware Aggregation): Detects structurally corrupted clients using a novel spectral fidelity metric derived from graph Laplacians, and dynamically reweights client contributions based on estimated corruption levels.
* RLSR (Robust Local Structure Reconstruction): Repairs corrupted local graph structures by aligning them with global model feature similarities, selectively pruning and reconnecting edges based on cosine similarity of node embeddings.

Extensive experiments on 7 benchmark datasets with synthetic noise under various structural noise levels show that FedSDR consistently outperforms the baseline algorithms.

**Strengths:**

1. The paper addresses an important and underexplored problem: structural noise in FGL. It highlights two practical challenges: 1) global inconsistency due to corrupted clients, and 2) local performance degradation on those clients.
2. The proposed spectral fidelity metric is grounded in Laplacian quadratic forms, enabling unsupervised, privacy-preserving detection of structural noise. The metric is efficiently computed locally and integrated into the FL aggregation via a principled reweighting scheme (SNAA).
3. RLSR leverages global model embeddings to detect and repair local structural corruption in a feature-aware manner. The joint pruning and reconnection based on cosine similarity helps maintain message-passing capacity while reducing noise impact.
4. The experiments cover a diverse set of datasets with varying graph types and structural noise settings, and include comprehensive comparisons against a wide range of baselines.

**Weaknesses:**

1. The introduction of SNAA and RLSR likely incurs additional computational and communication overhead (e.g., computing graph Laplacians, exchanging structural statistics, and constructing similarity matrices). However, these overheads are neither quantified nor discussed, which is critical in federated learning scenarios.
2. Although SNAA operates on local metrics and claims to preserve privacy, RLSR depends on embeddings derived from the global model, which may raise potential privacy concerns within the FL context.
3. RLSR's reliance on global similarity may cause over-regularization, especially when local client semantics differ from global trends (e.g., under non-IID features). There is limited discussion or evaluation on how FedSDR behaves when clean clients are semantically diverse.
4. Despite strong empirical results, the framework lacks formal convergence analysis or robustness guarantees, which would strengthen the theoretical foundation of the proposed method.
5. The experiments appear to evaluate only a single GNN architecture, which limits insight into the generalizability of FedSDR. Incorporating additional GNN models would provide a more comprehensive assessment of its robustness across architectures.

Minor issues:

1.  Abbreviations should be defined upon their first appearance in the paper, such as SNAA and RLSR in the abstract.
2. Figure 3 would be clearer with properly labeled x- and y-axes, which are currently missing.

**Questions:**

1. The paper does not provide details on the GNN model configurations used in the experiments. Could the authors clarify the specific architectural and training configurations of the GNN model?
2. Beyond the Louvain algorithm for graph partitioning, have the authors explored alternative partitioning methods? If so, how do the resulting performance and robustness compare across different strategies?

---

### Official Review · Reviewer_pe3T · 2025-11-01

**Soundness:** 2
**Presentation:** 2
**Contribution:** 1
**Rating:** 2
**Confidence:** 4

**Summary:**

This paper proposes FedSDR, a federated graph learning (FGL) framework to address structural noise via two components: Spectral-guided Structural Noise-Aware Aggregation (SNAA) for global client weighting and Robust Local Structure Reconstruction (RLSR) for local graph repair.

**Strengths:**

1. Identifies structural noise (e.g., missing/spurious edges) as an underaddressed FGL gap, matching real-world scenarios like fraud detection.
2. Tackles noise at global (aggregation) and local (structure) levels, aligning with FGL's multi-scale challenges.
3. Tests across diverse datasets (homophilic/heterophilic, small/large-scale) to validate robustness.

**Weaknesses:**

1. SNAA's metric is unreliable: It uses a spectral metric to label "corrupted" clients but fails to distinguish noise-induced spectral changes from intrinsic differences between graph types (e.g., Roman-empire vs. PubMed), leading to arbitrary weighting.
2. RLSR has circular logic: Relies on global model embeddings (trained on pre-repaired, corrupted data) as a "healthy" reference - no justification for this flawed baseline.
3. Missing critical baselines: Claims to outperform state-of-the-art but omits comparisons with structural noise-focused methods or their federated variants.
4. Noise design is unclear: Highlights adversarial noise (e.g., fraud) as a use case but doesn't specify if experiments use random or adversarial noise, eroding real-world relevance.
5. Privacy-utility tradeoffs ignored: Mentions differential privacy but provides no parameters (e.g., noise intensity) or accuracy impact - core to FGL's purpose.
6. No novelty: SNAA adapts existing robust FL ideas to spectral features; RLSR uses standard graph repair, with no FGL-specific innovations.

**Questions:**

1. How does SNAA's metric distinguish noise from intrinsic graph type differences?
2. Were experiments using random or adversarial noise? How does FedSDR perform on adversarial noise?
3. Why is the pre-repaired global model a valid reference for RLSR?
4. What differential privacy parameters were used, and how did they affect accuracy?
5. How does FedSDR compare to FedAvg + standalone graph repair?

---

### Note · Authors · 2025-11-12

I have read and agree with the venue's withdrawal policy on behalf of myself and my co-authors.